# Dispersal and Survival of Captive-Reared Threatened Fishes in a Tonle Sap Lake Reserve

Teresa Campbell [1],*, Peng Bun Ngor [2,3], Bunyeth Chan [2,3,4], Jackman C. Eschenroeder [5], Elizabeth Everest [6], Sudeep Chandra [6], Seila Chea [3], Kakada Pin [2,3,7], Samol Chhuoy [2,3,7], Soksan Chhorn [2,3], Sothearith Soem [2,3,8], Mecta Sup [2,3], Chheng Phen [9], Hoy Sreynov [10], Thay Somony [10], Chheana Chhut [3] and Zeb S. Hogan [6]

[1] McGinley and Associates, 5410 Longley Ln, Reno, NV 89511, USA
[2] Faculty of Fisheries, Royal University of Agriculture, Sangkat Dongkor, Khan Dongkor, P.O. Box 2696, Phnom Penh 120501, Cambodia
[3] Wonders of the Mekong Project, c/o Inland Fisheries Research and Development Institute (IFReDI), Fisheries Administration (FiA), No. 186, Preah Norodom Blvd., Khan Chamcar Morn, P.O. Box 582, Phnom Penh 12300, Cambodia
[4] Faculty of Agriculture, Svay Rieng University, National Road No.1, Svay Rieng 20306, Cambodia
[5] FISHBIO, 1617 S. Yosemite Ave, Oakdale, CA 95361, USA
[6] Department of Biology and Global Water Center, University of Nevada, 1664 N Virginia St, Reno, NV 89557, USA
[7] Department of Biology, Faculty of Science, Royal University of Phnom Penh, Phnom Penh 12156, Cambodia
[8] Centre for Biodiversity Conservation, Royal University of Phnom Penh, Russian Confederation Blvd, Phnom Penh 12156, Cambodia
[9] Fisheries Administration (FiA), No. 186, Preah Norodom Blvd., Khan Chamcar Morn, P.O. Box 582, Phnom Penh 12300, Cambodia
[10] Department of Aquaculture Development, Fisheries Administration (FiA), No. 186, Preah Norodom Blvd., Khan Chamcar Morn, P.O. Box 582, Phnom Penh 12300, Cambodia
* Correspondence: tcampbell@mcgin.com or tcampbs@gmail.com

**Abstract:** The Tonle Sap Lake in Cambodia supports several species of threatened megafish and contains one of the largest networks of freshwater fish reserves in the world. Despite these traits, this system remains understudied in terms of its utility for endangered fish conservation and restoration. This study was the first of a series of planned fish releases designed to test the effectiveness of conservation supplementation programs in the Tonle Sap Lake. In March 2022 (during the dry season), 1582 captive-reared fishes, including 1538 striped catfish *Pangasianodon hypopthalmus*, 42 giant barb *Catlocarpio siamensis*, and two Mekong giant catfish *Pangasianodon gigas*, were tagged and released into a 986-hectare fish reserve to assess post-release dispersal and survival. Brightly colored external tags with unique identification numbers were used to facilitate tag returns. A high-profile release event was held to raise awareness about the activity, bringing attention to the importance of fish reserves and endangered species conservation, and disseminating information about the research and tag return and reward program. This, in concert with other efforts, served to be an important education and outreach tool and increased tag return rates. We found that mortality from fishing was rapid and very high. Nineteen percent of released fishes were recaptured in the first 2 days after the fish release, and 46% were recaptured by day 83 after the release, indicating intense fishing pressure on the Tonle Sap Lake fisheries. Eighty percent of recaptured fishes were caught in stationary gill nets, most within 10 km of the release site. Fishing mortality rates were independent of fish size or source (pond-reared or cage-reared). Environmental DNA (eDNA) was found to be capable of detecting each of these species' presence in the water at the release site and could prove to be a useful tool for endangered species monitoring and restoration. Future research should explore alternative release timing, release location, and other methods of increasing post-release survival. Ultimately, underlying sources of mortality, especially fishing, will need to be addressed for conservation supplementation programs to succeed in the Tonle Sap Lake. Conservation supplementation should not be viewed as a substitute for more fundamental conservation measures, such as maintenance of environmental flows, preservation of ecological connectivity, and science-based fisheries management.

**Keywords:** endangered species restoration; native fish conservation; critically endangered; megafauna; tag and release; translocation; protected aquatic area; freshwater protected area; conservation aquaculture; conservation supplementation; fishing mortality; tropical lakes; Tonle Sap Lake

## 1. Introduction

Global freshwater biodiversity is in crisis [1,2], and freshwater megafauna have experienced some of the steepest population declines, including a 97% drop in the numbers of large-bodied Asian fishes [3]. Consequently, a large proportion of these species are classified as Endangered or Critically Endangered [4,5]. Aquatic megafauna serve important ecological roles, such as shaping ecosystems, transporting nutrients, and structuring communities [4]. Additionally, megafauna often holds a place of cultural importance to local communities [6]. As such, the conservation and restoration of these species would confer numerous benefits to ecosystems and human populations globally.

The Mekong River Basin is a hotspot for freshwater biodiversity [7–9] and has the third highest number of freshwater megafauna species (species that attain a minimum body weight of 30 kg), including a high number of fishes [10]. Many of these megafishes are Endangered or Critically Endangered [11,12]. The Mekong Basin in Cambodia still contains a relatively high number of threatened megafishes [13] as well as one of the largest networks of freshwater fish reserves in the world, with a total protected area of 600 km$^2$ in the Tonle Sap Lake [14]. Captive populations of many species of threatened fish exist in ponds and reservoirs throughout Cambodia and Thailand. These characteristics, in conjunction with a government climate that supports biodiversity conservation action and experimentation, make Cambodia an ideal place for endangered megafish restoration research.

Cambodia's Tonle Sap Lake, which is connected to the Mekong River via the Tonle Sap River, is a global biodiversity hotspot [15] and contains one of the largest freshwater fish reserves in the world [14]. Several species of megafish are known to have historically occurred in the Tonle Sap ecosystem [16–18]. Three of these are the striped catfish *Pangasianodon hypophthalmus* (Endangered), giant barb *Catlocarpio siamensis* (Critically Endangered), and Mekong giant catfish *Pangasianodon gigas* (Critically Endangered), all of which have high economic and cultural value.

The conservation impact of release (i.e., translocation) of captive-reared fish into the wild has been widely discussed due to the multiple rationales associated with the practice, including stocking to increase recreational fishing opportunities or boost commercial fisheries [19,20], release for ceremonial or religious purposes [21], or—as in the case of this study—as a potential conservation tool for restoration of endangered species populations [22]. Stocking to boost fishing or fisheries is most commonly criticized when non-native species are utilized and have been shown to have broad negative impacts on native species and aquatic ecosystems [23]. Likewise, release for ceremonial purposes has resulted in introductions of invasive species and caused human health concerns [24], but also, in some cases, has appeared to bolster populations of commercially important and native fish species [25]. In cases of conservation supplementation, previous studies have emphasized the need to consider implications on genetic diversity [26], post-release acclimation and behavior [27], and intraspecific competition [28] while also monitoring the fate of captive-reared animals to determine program effectiveness [29].

Fish tagging is a common research technique used for animal identification [30]. External tags can provide information about fish ecology, such as movement and dispersal patterns, and population-level data, such as estimates of fishing mortality [30,31]. External "t-bar" style tags (i.e., external vinyl-coated tags with a t-shaped anchor) offer a number of advantages for tag-and-release research: long retention times, ease of application, and high visibility. External tagging studies often rely on fishers to report tag returns and assume that all tagged fish are recognized and reported [30]. The effectiveness of the program depends on effective outreach to the fishing community and reporting of tagged fish [32]. Tag-and-

release efforts are often publicized through press releases and the media, and reporting rates can be increased by providing a reward for reporting recaptured, tagged fish [33].

We assessed the survival and dispersal behavior of captive-reared, translocated endangered fishes introduced into a large fish reserve within the Tonle Sap Lake. This study was planned as the first of a series of tag-and-release efforts, which ultimately will help understand if the reintroduction of captive-reared fish to the wild is a viable restoration tool for these species in this system. This work may also be used to inform conservation practices and policies, especially concerning the management of fish reserves, maintenance of critical habitat and ecological connectivity, and fisheries issues, such as assessment of fishing mortality and gear selectivity.

## 2. Materials and Methods

The study was conducted in March 2022 during the dry, low water season in a Tonle Sap Lake fish reserve within former fishing lot #4, Siem Reap Province, Cambodia (Figure 1). (The commercial fishing lots in the Tonle Sap Lake were abolished by the Royal Government of Cambodia in 2012 [34] and replaced with a community use system that also established a 600-km$^2$ network of fish reserves throughout the lake. For more detail on the former lot system and transition to community use areas and fish reserves, refer to Cooperman et al. [14].) The area of the fish reserve used in this study was 986 ha (Royal Government of Cambodia sub-decree #37 issued on 7 March 2012). The dry season was selected for the timing of the release because previous research has shown that fish congregate in fish reserves during low water periods [35,36]. A floating house was anchored within the fish reserve to serve as a research and enforcement station. Fish were held in tanks on the station or in boats adjacent to the station and were tagged and released from the station. The GPS coordinates of the station at the time of the study were 13.19162, 103.89609 (decimal degrees, datum = WGS84).

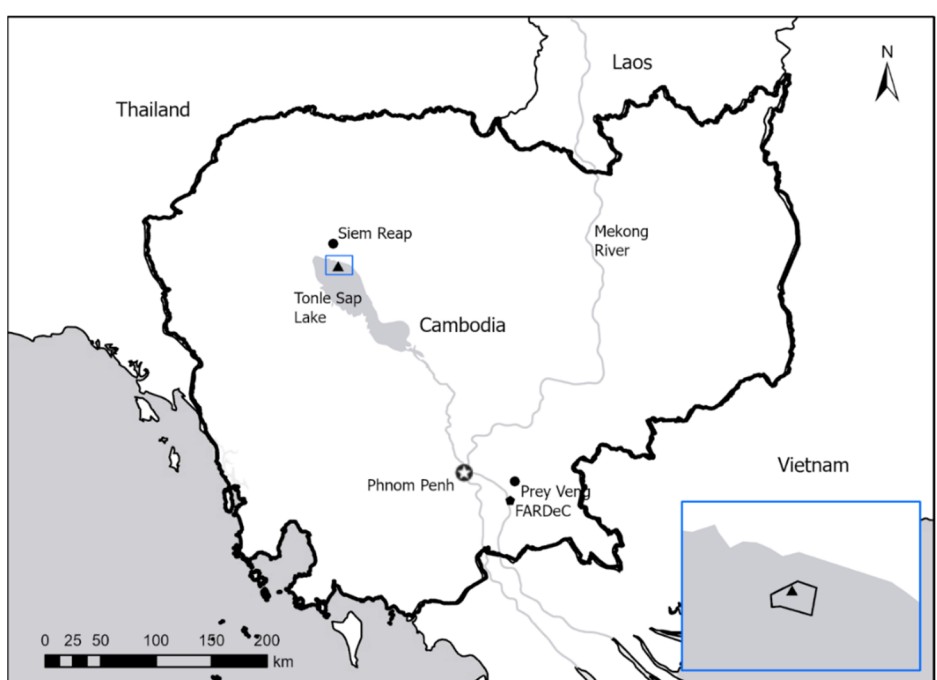

**Figure 1.** Map showing the location of the fish release (black triangle) within the Tonle Sap Lake, Cambodia. The black pentagon denotes the Freshwater Aquaculture Research and Development Center (FARDeC). The black polygon shown in the inset map delineates the boundary of the fish reserve in former fishing lot #4.

In anticipation of this study, over 6000 *P. hypophthalmus* individuals in 4-year classes (2017, 2019, 2020, 2021) were reared at the Freshwater Aquaculture Research and Devel-

opment Center (FARDeC) in Prey Veng Province, Cambodia, as part of a conservation supplementation program for endangered fishes. These fish were captured as larvae drifting down the Mekong River [37] and brought to FARDeC to be raised to larger sizes in ponds. FARDeC was also rearing some *C. siamensis* individuals purchased from small-scale vendors after being incidentally harvested as juveniles. Only a small number of *P. gigas* individuals were being reared at FARDeC because this fish is exceedingly rare. A subset of these fish from FARDeC was used for the tag and release study (see below). Additional *P. hypophthalmus* individuals were sourced from local fishers living along the Tonle Sap Lake in Siem Reap Province. Fish sourced from fishers had been caught as juveniles and held in cages in the lake to grow to larger sizes (which fetch a higher market sale price).

Fish from FARDeC were transported by truck on the night of 2 March 2022 to the Toeuk Vel Aquaculture Station in Siem Reap, where they were held in tanks until they were transported to the fish reserve in the early hours of March 4 while it was still dark and then released in the morning of March 4. Fish sourced from fishers were acquired and tagged on March 3–5 and released into the fish reserve on March 4–5. Fish were transported to the floating research station in boats lined with tarpaulin and filled with water.

*P. hypophthalmus* individuals were tagged with T-bar tags, and *C. siamensis* and *P. gigas* individuals were tagged with disc tags, which are better for larger fish and those with large scales, and also have information printed on them, informing fishers what to do when the tags are found (Figure 2). Each tag had a unique identification number and was brightly colored for easy identification by fishers. Tags were placed into the dorsal musculature near the dorsal fin [38]. The fish's total length was measured to the nearest millimeter during tagging. Handling time was minimized to reduce tagging-associated stress and mortality. This work was done under Institutional Animal Care and Use Committee (IACUC) protocol 20-10-1098-1.

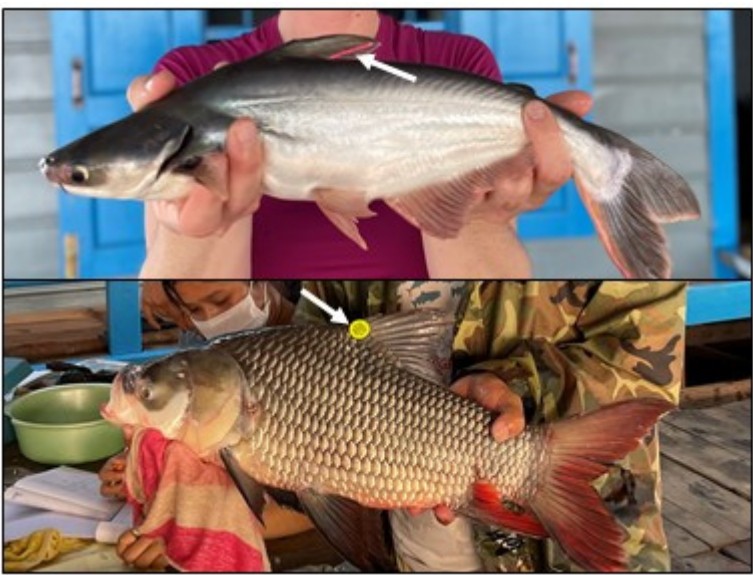

**Figure 2.** Types of tags used in the study. *(Top)* T-bar tag on a *Pangasianodon hypophthalmus*. *(Bottom)* Disc tag on a *Catlocarpio siamensis*.

Most of the fish were released during a well-publicized fish release event on March 4. Representatives of the Cambodian Fisheries Administration, including the Director General, attended and spoke during the fish release. The release was covered by multiple governments and independent news outlets, including the Agence Kampuchea Presse, National Television of Cambodia, Cambodian News Channel, Bayon TV, Hang Meas TV, ThmeyThmey and ThmeyThmey25, the Khmer Times, Fresh News Asia, and Southeast Asia Globe.

The media coverage at the fish release event was one of several ways that fishers were informed about the tagged fish and the reward for returning the tags to their Provincial Fisheries officers (who in turn would call us and give us the tag information). This information was also disseminated to the fishers in the two days following the fish release by distributing posters to local authorities and fishing communities and also by word of mouth through local communities. Additionally, in the weeks following the fish release, project scientists stayed in regular contact with the management of the Siem Reap Fisheries Administration Cantonment from provincial to communal levels, as well as commune heads in the area. Fishers were notified after the release in order to minimize the chance of increased fishing pressure around the fish reserve as a result of fishers knowing about the release.

Fishers received a reward of 10,000 Riel (2.50 USD) for each tag returned. The reward was not used to incentivize people to fish; people normally fish every day in the Tonle Sap Lake. The purpose of the reward was to encourage fishers to return the tags and compensate them for their time and effort. The information collected with each tag return included the tag number, date of capture, location of capture, and the type of fishing gear with which the fish was captured.

The tag and release research was paired with environmental DNA (eDNA) sampling to see if eDNA could detect the presence of these species within the fish reserve and potentially be used as a monitoring tool within Tonle Sap reserves. Samples were collected with single-use eDNA collection kits containing a 5 μm filter (Jonah Ventures, Boulder, CO, USA), and collected filters were preserved with Longmire's solution to stabilize captured DNA and then sealed with sterile plastic caps to prevent contamination during transport to the laboratory. Samples from the lake were collected from the surface, and an effort was made to filter as large a volume as possible using the 60 mL syringe contained within each sample kit. Due to high turbidity, it was only possible to filter relatively small volumes of water (13–60 mL) before the filter became clogged with sediment and other particles.

A total of seven were collected over a time period ranging from three days prior to the release to 57 days post-release (Table 1). One eDNA sample was collected from the fish reserve on March 1 (three days prior to the fish release). During the release, samples were collected from the holding tanks of each of the three species to serve as positive controls and verify the ability of the eDNA methodology to detect each of the species. Two samples were collected from the fish reserve on March 5 (one day after the fish release). A final sample was collected on April 29 (57 days after the release). An effort was made to collect the samples from the lake in the same location on the tagging platform.

**Table 1.** The time, location, and volume of water filtered for each of the eDNA samples.

| Sample | Date | Time | Location | Total Volume (mL) |
|:---:|:---:|:---:|:---:|:---:|
| 1 | 1 March 2022 | 15:20 | Lot 4, Tonle Sap Lake | 60 |
| 2 | 3 March 2022 | 16:45 | *P. hypophthalmus* tank | 13 |
| 3 | 4 March 2022 | 8:50 | *C. siamensis* tank | 25 |
| 4 | 4 March 2022 | 9:05 | *P. gigas* tank | 65 |
| 5 | 5 March 2022 | 14:20 | Lot 4, Tonle Sap Lake | 50 |
| 6 | 5 March 2022 | 14:25 | Lot 4, Tonle Sap Lake | 25 |
| 7 | 29 April 2022 | 12:48 | Lot 4, Tonle Sap Lake | 60 |

Samples were shipped to Jonah Ventures for metabarcoding analysis using MiFish primers [39]. These primers target the mitochondrial 12 S ribosomal RNA (rRNA) gene and have been shown to have good discriminatory power for the identification of fish families, genera, and species [40]. Detailed laboratory methodology is provided in the Supplementary Materials.

### 3. Results

A total of 1582 fish were tagged and released (Table 2). This included 1538 *P. hypophthalmus*, 42 *C. siamensis*, and two *P. gigas*. Of the 1538 *P. hypophthalmus*, 832 were sourced from FARDeC (pond-reared) and 706 from fishers on the Tonle Sap Lake (wild/cage-reared). The *C. siamensis* and *P. gigas* individuals were all sourced from FARDeC.

**Table 2.** Summary of the numbers of fish released and recaptured. TSL indicates Tonle Sap Lake.

| Species | Number Released | Number Recaptured |
|---|---|---|
| *P. hypophthalmus* | 1538 (832 FARDeC, 706 TSL) | 706 (345 FARDeC, 361 TSL) |
| *C. siamensis* | 42 | 26 |
| *P. gigas* | 2 | 0 |
| **TOTAL** | **1582** | **732** |

The first returned tags were reported on March 5, one day after the fish release event. One hundred twenty-four tags (8% of the total released) were reported on March 5, followed by 175 (11%) on March 6. This came to a total of 19% of fish recaptured in the first two days after the release.

In total, 732 tags (46% of the total released) were returned by 26 May 2022, 83 days after the release (Table 2). Seven hundred six were *P. hypopthalmus* (46% of the total *P. hypophthalmus* released), and 26 were *C. siamensis* (62% of the total *C. siamensis* released). Three-hundred forty-five (49%) of the returned *P. hypopthalmus* tags were from FARDeC, and 361 (51%) were from the Tonle Sap Lake. This represented 41% of the released *P. hypophthalmus* from FARDeC and 51% of the released *P. hypophthalmus* from the Tonle Sap Lake.

Five hundred eighty-two (80%) of the recaptured fish were caught in stationary gill nets, and 145 (20%) were caught in arrow-shaped traps. No gear was reported for five of the recaptured fish.

Fish body size was similar between the population of tagged fish and the population of recaptured fish (Table 3).

**Table 3.** Summary of fish body sizes (total length) for tagged and recaptured fish. Total length is reported in cm.

| Species | Statistic | Tagged Fish | Recaptured Fish |
|---|---|---|---|
| *P. hypophthalmus* | Minimum | 13.0 | 18.0 |
| | Mean (SD) | 32.7 (9.3) | 34.1 (8.9) |
| | Maximum | 68.5 | 68.5 |
| *C. siamensis* | Minimum | 30.0 | 30.0 |
| | Mean (SD) | 42.6 (11.0) | 42.0 (11.0) |
| | Maximum | 68.5 | 67.0 |

Most fish were captured close to the release point (within approximately 10 km) outside of the fish reserve in Siem Reap Province. The longest distance traveled by a recaptured fish was approximately 56 km. This fish traveled southeast to the mouth of the Pursat River in Pursat Province. This fish was a 40.0 cm *P. hypopthalmus* sourced from the Tonle Sap Lake. It was recaptured on March 19, 2022, 15 days after its release on March 4.

Analysis of the eDNA samples resulted in positive detections of DNA belonging to each of the three study species, plus one additional species (Borneo river sprat *Clupeoides borneensis*) as well as several sequences that could only be identified at the family level (Table 4).

**Table 4.** The taxa detected in each of the seven eDNA samples.

| Sample | Days after Release | Taxa Detected | Sample Location |
|---|---|---|---|
| 1 | −3 | * Cypriniformes Cyprinidae *Catlocarpio siamensis* | Lot 4, Tonle Sap Lake |
| 2 | 0 | * Siluriformes Pangasiidae *Pangasianodon hypophthalmus*<br>Siluriformes Pangasiidae—genus and species unknown | *P. hypophthalmus* tank |
| 3 | 0 | * Cypriniformes Cyprinidae *Catlocarpio siamensis*<br>*Siluriformes Pangasiidae *Pangasianodon hypophthalmus*<br>Siluriformes Pangasiidae—genus and species unknown | *C. siamensis* tank |
| 4 | 0 | * Cypriniformes Cyprinidae *Catlocarpio siamensis*<br>Cypriniformes Cyprinidae—genus and species unknown<br>* Siluriformes Pangasiidae *Pangasianodon gigas*<br>* Siluriformes Pangasiidae *Pangasianodon hypophthalmus*<br>Siluriformes Pangasiidae—genus and species unknown | *P. gigas* tank |
| 5 | 1 | Clupeiformes Clupeidae *Clupeoides borneensis*<br>* Cypriniformes Cyprinidae *Catlocarpio siamensis*<br>* Siluriformes Pangasiidae *Pangasianodon gigas*<br>*Siluriformes Pangasiidae *Pangasianodon hypophthalmus*<br>Siluriformes Pangasiidae—genus and species unknown | Lot 4, Tonle Sap Lake |
| 6 | 1 | * Siluriformes Pangasiidae *Pangasianodon gigas*<br>Siluriformes Pangasiidae—genus and species unknown | Lot 4, Tonle Sap Lake |
| 7 | 57 | No fish taxa detected | Lot 4, Tonle Sap Lake |

Note: * indicate study species.

Notably, each of the samples collected from the holding tanks prior to fish release (Samples 2–4) resulted in the detection of DNA belonging to the respective species housed therein. The sample collected three days prior to the release (Sample 1) detected only *C. siamensis*, indicating that this species may have been present in the vicinity of the release site prior to the release of the tagged individuals. The sample collected one day after fish release (Sample 5) was found to contain DNA from all three released species; the paired sample (Sample 6) only detected *P. gigas*. No fish taxa were detected in the sample collected 57 days after release (Sample 7).

## 4. Discussion

Fish reserves and conservation supplementation can be effective approaches to increasing populations of threatened animals [41,42]. Many factors impact program success, which is often measured in terms of post-release survival and reproduction. Factors of success include, but are not limited to, reserve design and management, release location and timing, species selection, fish age and origin, habitat availability, engagement of local stakeholders, and addressing underlying threats [42–46]. While other Mekong studies have examined wild fish migration [47] and quantified recapture rates of tagged fish [48], this study was the first large-scale tag-and-release of multiple species of endangered and critically endangered fishes into a fish reserve in the Tonle Sap Lake to assess their survival and dispersal. The high recapture rates indicate extreme fishing pressure around the reserve and the high vulnerability of fish during the low water period, but they also demonstrate sufficient community outreach and cooperation from local fishers to track movement and survival. While more study is needed, captive-reared fish often behave differently than wild fish [49–51], and as such, the use of pond and cage-reared fish may have contributed to the rapid dispersal outside of the reserve and the corresponding high recapture rate. The results of this work have implications for future research and conservation efforts, especially actions associated with endangered Mekong fish species, Tonle Sap fish reserves, and captive-reared fish behavior, dispersal, and survival.

Although no fishing was observed inside the reserve, intense fishing pressure has been well documented in the Tonle Sap Lake [18,52], and extensive use of gill nets and

arrow-shaped traps outside the reserve led to a high recapture rate within a short period of time. The high recapture rate also showed that fish quickly left the reserve after their release. If the reserve is to be utilized to protect or supplement populations of highly mobile fish, then strategies would need to be implemented to maximize survival, such as changing the timing of releases to coincide with seasonal fishing closures; releasing fish during the wet season when fish are more difficult to catch; acclimating fish to the lake prior to release [53]; ensuring the release location is in preferred habitat [54]; or developing a connected reserve network [35] that protects fish during their likely long-distance spawning migrations and post-spawning dispersal, a behavior that has been documented in related species [55]. Future research may also compare recapture rates between migratory and non-migratory species. The species used in this study were all highly migratory, which may have contributed to the high recapture rates. While the goal is to minimize post-release mortality, susceptibility to fishing and other forms of mortality should be expected [46,54,56] and therefore considered in species selection, release location, and release timing in future studies.

Previous studies have indicated that fish origin, translocation method, and rearing technique influence fish behavior and survival [57]. In our study, however, fish origin and rearing methods did not appear to influence post-release behavior or fishing mortality. Similar proportions of returned *P. hypophthalmus* originated from FARDeC (pond-reared) and the Tonle Sap Lake (wild/cage-reared), with 345 (49%) and 361 (51%) returns, respectively. This equated to 41% of the released FARDeC fish and 51% of the released Tonle Sap Lake fish. This result is somewhat surprising because, during tagging, the Tonle Sap Lake fish appeared to be in better condition than the FARDeC fish, which was likely the result of many hours spent traveling from FARDeC to the reserve, which is both physiologically and physically stressful for fish [58,59]. Nonetheless, it appeared that pond-reared and cage-reared fish exhibited similar dispersal behavior and that fishing pressure was high enough that all fish had an equally high probability of capture regardless of origin or rearing method. It would be informative to tag and release wild fish captured from the fish reserve itself to determine whether or not a focus on improving the protection of existing wild stocks is a more effective long-term conservation strategy. However, as it is currently very difficult to find our study species in the wild, other species may need to be used for this work.

Although many types of fishing gear are used in the Tonle Sap Lake, the majority (82%) of our recaptured fish were captured in gill nets. As has been shown in marine systems, certain fishing gears can have larger negative impacts on populations of endangered and migratory fish than other gears, and identifying particularly damaging gears can guide management decisions [60–63]. Regulating the use of gill nets (e.g., by season, fishing site, and length and mesh sizes) may be an effective way to contribute to endangered species conservation. Large-mesh gill nets are already banned in some areas of the Cambodian Mekong to protect stocks of spawning fish [64]. Temporary closures on the use of gill nets, or restrictions on net length, mesh size, or fishing location may allow endangered fishes to avoid some mortality while still allowing fishers to fish with other gears. Furthermore, there is currently a regulation banning fishing activities within a few hundred meters of the border of the fish reserve, which may need to be more effectively enforced. The high catch rate in our study can be partially attributed to gill nets set just outside of the fish reserve. Ultimately, addressing the underlying threats to these populations, especially high fishing pressure, will be critical to successful restoration [65].

Study results revealed that eDNA monitoring could serve as a rapid, inexpensive, and efficient tool for monitoring endangered species restoration programs in the Tonle Sap Lake and other parts of the Mekong. Despite challenges with poor taxonomic resolution for several of the detected DNA sequences, the analysis methods used here were demonstrated to effectively detect all three of the study species in the lake. *C. siamensis* was detected in the sample collected 3 days prior to release, indicating that this species may have been present in the vicinity of the release site prior to the release. The sample collected 1 day

after the release contained DNA from all three released species. In a system as large as the Tonle Sap, where capturing rare fish is difficult, eDNA can provide a way to monitor the distribution of species and confirm species presence and may potentially be used as a guide for targeted sampling with traditional methodologies [66]. Further, the collection of eDNA samples requires minimal training and can be easily conducted across large geospatial areas and across seasons [67]. As genetic reference libraries are expanded, the resolution of metabarcoding approaches will continue to improve, and the incorporation of eDNA sampling into fisheries monitoring will become increasingly valuable [68].

However, the results of this study also demonstrate the remaining challenges related to the successful application of eDNA metabarcoding for fisheries monitoring. For example, the identification of detected DNA sequences to species or even genus level was not possible in all cases. This lack of resolution is in part due to a lack of available reference sequences for many of the species that occur in the Mekong Basin [68]. Moreover, many of the families and genera in the Mekong Basin are represented by a very high diversity of species, and, in many cases, there is no species specificity within or even between genera for the targeted region of the mitochondrial genome [68].

Additionally, the only species whose DNA was detected in the sample collected 57 days post-release was the domestic pig (*Sus scrofa*). The failure to detect any fish taxa in this sample may have resulted from challenges inherent to metabarcoding, namely the potential for preferential amplification of certain target sequences. This bias in amplification may occur because DNA from a particular species is abundant at a sample site or because the template sequences from a given species have fewer mismatches with the primers than those of other species. These biases in the amplification of template DNA can lead to false negatives (i.e., the failure to detect a species even when its DNA is present in the sample). Because budgetary and logistical constraints prevented eDNA sample replication for this study, it is not possible to comprehensively evaluate potential causes for the lack of fish taxa detected in this single sample, but the amplification of only DNA belonging to *Sus scrofa* may be the result of an abundance of that species' DNA at the location where the sample was collected. Subsequent analysis with a different pair of metabarcoding primers may improve the resolution of this sample.

Community outreach was essential for raising awareness about the project rationale and our tag return and reward program and for obtaining recapture results [69]. Multiple forms of outreach were used to inform fishers about the tag return and reward program, including local media, communication with local authorities, poster distribution, and community visits. Feedback from fishers indicated that social media, such as Facebook and Telegram, and communication through their social networks were important methods of disseminating information about the program. Fishers also observed their peers returning tags for rewards, which encouraged them to do likewise. There was good cooperation with the project among fishers, which seemed to indicate that the reward amount was sufficient to encourage them to make the effort to return tags. However, direct engagement with a few local fishers may have been even more important for gaining fisher participation as these fishers were able to communicate to others in ways that they could understand and alleviate fears of communicating to authorities. There was also good cooperation from local community leaders and authorities.

This activity served as an education and outreach opportunity to raise awareness about the status of endangered species, an essential action for highlighting conservation issues among the general public [69]. The stakeholders reached with these messages included local fishers, community leaders, members of law enforcement, and public officials. In addition, several university students participated in the research and outreach associated with this event, providing many of them with their first field research experience. The release event drew attention to the problems of overfishing and the importance of fish reserves. Following the release event, the prime minister called for increased enforcement of fisheries regulations and a reduction of illegal fishing in the Tonle Sap Lake [70].

Other key strengths of this study approach were its financial sustainability and ease of implementation. The fish used in this study were purchased for a relatively low price, which means follow-up studies are financially feasible. External tagging and monitoring are simple and straightforward, making it easier to find people to do the work and facilitate fisher participation. Good relationships between researchers, universities, the Fisheries Administration, and local community leaders are already in place and can be leveraged for future collaborative efforts. The strong partnership with the Fisheries Administration facilitated all aspects of the work, including making local contacts, following up with fishers, and providing aquaculture facilities for rearing or holding fish. Furthermore, the existing fish reserve system provides an ideal opportunity for conservation research. All of these factors can facilitate similar future studies.

Tagged fish experienced high mortality rates shortly after the release, indicating that conservation supplementation faces challenges as a viable method for species conservation in the Tonle Sap Lake. This study reinforces previous research highlighting the benefits of scientifically-based guidelines and protocols for future reintroductions, including incorporating life history and genetics, as well as the design of a post-release monitoring program [29,71]. Most successful reintroduction programs take place over multiple years and employ multifaceted approaches (e.g., habitat restoration, stakeholder involvement, etc.), and may take several trials to develop a successful approach [42]. Thus, follow-up studies are planned to test this approach under conditions that may be more favorable for survival, such as the timing of the release during the wet season and comparing recapture rates among captive-reared and wild fish captured inside the reserve network. As these fish are all highly migratory, future research may also consider these species' seasonal migration corridors and other critical habitats apart from those in the lake. Given the high fishing mortality and dispersal patterns observed in this study, conservation supplementation should not be a substitute for more fundamental conservation measures, such as preservation of migration corridors, protection of critical habitats, and maintenance of the seasonal flood pulse, all of which are essential to the long-term survival of the three species included in this study [17,37,72].

**Supplementary Materials:** The following supporting information can be downloaded at: https://www.mdpi.com/article/10.3390/w14192995/s1, eDNA Laboratory Methodology [73–78].

**Author Contributions:** Conceptualization, T.C. and Z.S.H.; methodology, T.C., Z.S.H. and S.C. (Seila Chea); formal analysis, T.C., J.C.E.; investigation, T.C., Z.S.H., P.B.N., B.C., E.E., S.C. (Seila Chea), K.P., S.C. (Samol Chhuoy), S.C. (Soksan Chhorn), S.S., M.S., C.C.; resources, Z.S.H., C.P., H.S., T.S.; data curation, T.C., E.E., S.S., M.S.; writing—original draft preparation, T.C.; writing—review and editing, T.C., Z.S.H., P.B.N., B.C., J.C.E., E.E., S.C. (Sudeep Chandra), S.C. (Seila Chea), K.P., S.C. (Samol Chhuoy), S.C. (Soksan Chhorn), S.S., M.S., C.P., H.S., T.S., C.C.; visualization, T.C.; supervision, Z.S.H., S.C. (Seila Chea); project administration, Z.S.H., S.C. (Seila Chea); funding acquisition, Z.S.H. All authors have read and agreed to the published version of the manuscript.

**Funding:** This research was funded through the United States Agency for International Development (USAID) 'Wonders of the Mekong' Cooperative Agreement No: AID-OAA-A-16-00057.

**Institutional Review Board Statement:** Research protocols were approved by the University of Nevada, Reno, International Animal Care and Use Committee (IACUC protocol ID 20-10-1098).

**Informed Consent Statement:** Not applicable.

**Data Availability Statement:** Data are available upon request.

**Acknowledgments:** We particularly thank the fishers and Fisheries Administration officers, as well as Srey Keo Sopheak, of the Siem Reap Fisheries Administration Cantonment, Siem Reap Province, Cambodia, who gave us their time and kind participation in the study. We express gratitude to the Director General of the Cambodian Fisheries Administration, His Excellency Poum Sotha, for hosting the fish release event. Many thanks to Aaron Koning, University of Nevada, Reno, for providing constructive comments on the manuscript. This project would also not have been possible without the work of several people: Em Thearith, Vice Chief of Aquatic Health and Disease Management

Office of the Department of Aquaculture Development, monitored fish health at FARDeC and helped prepare the fish for safe transport to the release site; Meng Sothai, Director of FARDeC, provided the rearing ponds for the fish used in the study as well as general project support; and Ros Narin, Officer of FARDeC, cared for the fish in the rearing ponds. We also thank Thach Phanara for his assistance in capturing the fishes reared at FARDeC, transporting fish to the release site, and tagging fish during the event. Finally, we thank Bunthang Touch, Acting Director of IFReDI, for his administrative facilitation of the fish release.

**Conflicts of Interest:** The authors declare no conflict of interest.

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
