# Peer review of "Dispersal and Survival of Captive-Reared Threatened Fishes in a Tonle Sap Lake Reserve"

_water, doi:10.3390/w14192995_

Round 1

Reviewer 1 Report

Dear authors,

Congratulations for the results of your work.

In my opinion the paper can be improved if the citations/references will be enriched especially in the first part of the paper.

Please be sure that all the material and methods information belong there and not to the introduction part.

Also, I think is needed to highlight in a more complex/integrated/applied way the importance of your research results to all the Mekong basin fish fauna assessment, monitoring and management.

I think that the discussion about the importance of the natural and anthropogenic factors which influence the dispersion gradient and the rate of survival can be improved/enriched. 

Success and all the best!

Reviewer 2 Report

Please find my comments in the attached file. The main concern about the paper is the way of writing. I believe author can improve the MS's  readability. 

Reviewer 3 Report

Overall, this is a clear, concise, and well-written manuscript. The introduction is relevant to the studied issue. The materials & methods are appropriate, and detailed information regarding the tagging of fish before release and the contribution of the government sector and community in the released and post-released recapture of the fish is available. Overall, the results are clear, however why both wild and captive-reared fish showed the non-significant recaptured data need some clarification

Round 2

Reviewer 1 Report

Congratulations for the work and paper.

Reviewer 2 Report

The authors have resolved the issues raised in earlier version. But I want the author should check the English of the manuscript one more round specially the tenses. Good luck.